# Investigating the role of modality and training objective on representational alignment between transformers and the brain

**Hyewon Willow Han**[*1,2,3]      **Ruchira Dhar**[*1,4]      **Qingqing Yang**[*1,5]
hhan228@uwo.ca           rudh@di.ku.dk        yang.6118@osu.edu

**Maryam Hoseini Behbahani**[1]      **María Alejandra Martínez**[1,6]
maryam.hoseini2101@gmail.com          mm13852@nyu.edu

**Tolulope Oladele**[1,7]                    **Diana C. Dima**[2,3]
toladele@unimed.edu.ng                  ddima@uwo.ca

**Hsin-Hung Li**[†5]           **Anders Søgaard**[†4]         **Yalda Mohsenzadeh**[†1,2,3]
li.14492@osu.edu            soegaard@di.ku.dk          ymohsenz@uwo.ca

[1]Neuromatch Academy   [2]Western University   [3]Vector Institute   [4]University of Copenhagen
[5]The Ohio State University   [6]New York University   [7]University of Medical Sciences, Ondo

## Abstract

The remarkable performance of transformer models in both linguistic and real-world reasoning tasks coupled with their ubiquitous use has prompted much research on their alignment with brain activations. However, there remain some unanswered questions: what aspects of these models lead to representational alignment-the input modality or the training objective? Moreover, is the alignment limited to modality-specialized brain regions, or can representations align with brain regions involved in higher cognitive functions? To address this, we analyze the representations of different transformer architectures, including text-based and vision-based language models, and compare them with neural representations across multiple brain regions obtained during a visual processing task. Our findings reveal that both training data modality and training objective are important in determining alignment, and that models align with neural representations within and beyond the modality-specific regions. Additionally, the training modality and objectives seem to have an impact on alignment quality as we progress through the layers, suggesting that multimodal data along with a predictive processing objective may confer superior representational capabilities compared to other training objectives.

## 1 Introduction

The recent introduction of the transformer architecture [1], combined with a predictive processing training objective, has led to the rise of models that have achieved unprecedented performance in the domain of natural language processing. Nowadays, larger variants of these language models, known as large language models (LLMs) or multimodal language models (MLMs), have been shown to have superior language understanding and generation capabilities [2, 3], structured understanding of language [4–7], and significant performance in broader cognitive domain tasks like general reasoning [8–12] and planning [13, 14]. This has also led to a proliferation of research with transformers within

---

[*]These authors contributed equally to this work.
[†]Co-corresponding authors.

the neuroconnectionist research programme [15], focusing on alignment of their representations with brain activations [16–19], and there has been an increasing interest in several questions about their relation to human cognitive processes and how these models process and represent information compared to the human brain.

When it comes to humans, there has always been evidence of predictive processing playing an important role during language comprehension [20–22]. Predictive processing transformers today have shown superior capabilities in language processing - popular text-trained models like `Llama3-8B` [23] have shown robust language generation capabilities while those further trained on code like `CodeLlama-7B` [24] have also been shown to develop even advanced reasoning capabilities [25]. Moreover, the embeddings of transformer-based LMs have been shown to be robust at reflecting human judgements across language and vision inputs [26, 27]. This has led to the consideration of such transformer-based LMs as possible models of human language processing. However, while there is some research on determining pressures which impact such model alignment capabilities [28–30], there has been little work that specifically focuses on transformers and delineating design choices in them that improve alignment. Previous research has investigated the role of different architectures, task objectives and training diets on the alignment of biological and artificial systems [31, 32]. However, most of these studies have predominantly focused on visual processing and image-based tasks [33, 34], with relatively few exploring the multi-modal capabilities of artificial neural network models. On a related note, research has also shown that the human brain is highly modular: for example, linguistic and non-linguistic tasks are clearly separated from one another in the brain [35–38]. Recent work on the interface of LLMs and human language processing has also emphasized the need to separate language and general cognition [39, 40]. How valid is this domain specificity in the case of model representations?

These considerations lead us to our two central questions:

- Do input modality and training objective impact the representational alignment of transformers with the human brain?

- Can task or domain-specific representations from models align to brain regions with higher cognitive functions beyond modality-specialized regions?

To answer these, we consider the domain of visual processing tasks and compare representations from various deep generative models with brain activations across different regions. The stimuli and functional Magnetic Resonance Imaging (fMRI) data are taken from the BOLD Moments Dataset (BMD) [41], which contains fMRI data from 10 subjects viewing short natural videos. To study the impacts of training data modalities and objectives, we use six different types of models in our research: a convolutional neural network baseline model and five transformer architectures with varying input modalities and training objectives. Specifically, we use an image model (`ResNet-50`), a video model (`ViViTB`), a language model (`Llama3-8B`), a language model which was trained on programming languages (`CodeLlama-7B`), an image-language model (`BLIP-L`), and a video-language model (`LLaVA-OV-7B`). The code-trained language model was chosen because of the recent finding on its different behaviour on reasoning tasks compared to its natural text-based counterpart, `Llama-3-8B-Instruct` [11, 25]. We extract the models' hidden representations of the stimuli from their early, middle, and last layers, and apply representational similarity analysis (RSA) across 20 different human brain regions of interest (ROIs). Additionally, we apply searchlight RSA [42] to map where the model best reflects the local neural activation patterns.

Our results indicate that a combination of multimodal data and generalized predictive processing i.e. next-word prediction training objective is critical in improving the alignment with neural representations, the influence growing more pronounced as we hierarchically ascend model layers. This alignment also manifests for higher-level regions, highlighting the broad scope of representational convergence. This observation indicates that predictive processing, combined with multimodal data, may endow models with a more sophisticated and nuanced representational alignment capacity when compared to other training paradigms. Such results have considerable implications for research surrounding the cognitive capabilities of models and their ability to emulate human cognition.

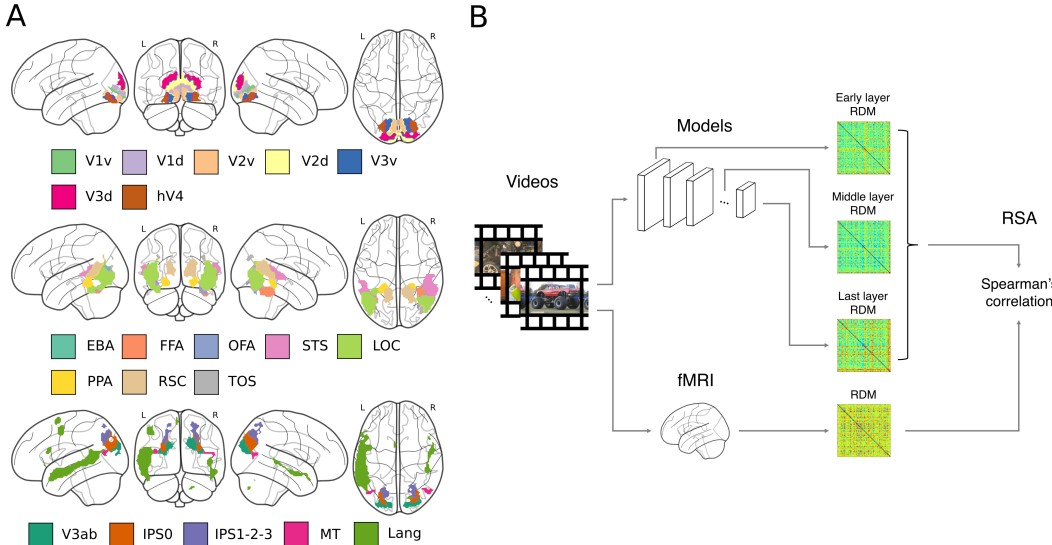

Figure 1: (**A**) Definitions of regions of interest (ROIs). Subject 1 was used for the visualization. (**B**) A schematic diagram of the comprehensive workflow for the analysis. For the visualization, `LLaVA-OV-7B` was used for model RDMs and V1v from Subject 1 was used for the fMRI RDM.

## 2 Methods

### 2.1 Stimuli

The stimuli used in our study consists of 102 short videos from the BOLD Moments Dataset (BMD) [41]. The videos presented to participants did not contain audio information or captions. In the scanner, subjects are instructed to fixate at the central fixation cross. Each video has a duration of 3 seconds and a frame rate between 15 and 30 frames per second, ensuring a diverse range of temporal resolutions. This test dataset is a carefully selected subset from the larger Memento10k dataset [43], which itself is derived from the extensive Moments in Time dataset [44] and its extended Multi-Moments in Time dataset [45]. The Memento10k dataset includes a broad spectrum of real-world activities and scenes, providing a rich context for evaluating vision-based models.

### 2.2 fMRI representations

BMD comprises data from 10 healthy subjects, with an average age of 27.01 years (SD = 3.96). Each subject participated in five fMRI sessions, including anatomical scans, functional localizer scans, and visual task fMRI sessions. During the main task, subjects viewed each video from the test set 10 times. They were instructed to maintain fixation on a central point throughout each video. Each video lasted 3 seconds and was followed by a 1-second interval.

We utilize 15 early and ventral visual ROIs defined from the localizer experiments conducted for each subject. These ROIs include early or mid-visual areas (V1v, V1d, V2v, V2d, V3v, V3d, hV4), which are critical for processing basic visual features. Additionally, body-selective regions (EBA), object-selective regions (LOC), face-selective regions (FFA, OFA, STS), and scene-selective regions (PPA, RSC, TOS) were included to capture specialized visual processing. The number of voxels for each ROI and each subject was capped at 1000, following the procedure described in the BMD paper, where the top 1000 ROIs were limited by masking each subject's FWE corrected t-contrast map with the corresponding binarized t-contrast probability map [41]. This approach allows for a detailed investigation into how different visual areas respond to the diverse stimuli presented in the videos.

Additionally, 3 dorsal visual stream areas defined from anatomical landmarks in BMD were adopted, including V3ab, IPS0, and IPS1-2-3 defined with a maximum probability map [46]. The middle temporal visual area (MT) was also extracted [47], as a part of the visual motion processing pathway [48, 49]. The language area (inferior frontal gyrus, inferior frontal gyrus orbital, middle frontal

Table 1: Detailed overview of model specifications

| Model | Architecture | Modality | Training objective | Number of parameters | Used layers (early, middle, last) |
|---|---|---|---|---|---|
| ResNet-50 | Convolutional Neural Network | Image | Classification | 25M | 1, 2, 4 |
| ViViT-B | Transformer | Image | Classification | 89M | 3, 5, 12 |
| Llama3-8B | Transformer | Natural Language | Predictive Processing | 8B | 8, 15, 32 |
| CodeLlama-7B | Transformer | Natural Language, Code | Predictive Processing | 7B | 8, 15, 32 |
| BLIP-L | Transformer | Image, Natural Language | Image Captioning | 470M | 6, 11, 24 |
| LLaVA-OV-7B | Transformer | Video, Natural Language | Predictive Processing | 7B | 7, 13, 28 |

gyrus, posterior temporal region and anterior temporal region) was extracted for each individual with a probabilistic atlas, LanA [50]. In conclusion, 20 ROIs were included in the current study. The visualization of all 20 ROIs is shown in Figure 1A.

## 2.3 Model representations

To measure the impact of model design choices (input modality and training objective), we consider a suite of six models, including a baseline model: ResNet-50 (convolutional neural net for image classification) [51], ViViT-B (video-vision transformer for classification) [52], Llama3-8B (a typical autoregressive language model) [23], CodeLlama-7B (LLAMA-variant further trained on code) [24], BLIP-L (a vision language model) [53], and LLaVa-OneVison-7B (a video language model) [54]. Refer to Table 1 for a more detailed overview of their architecture and training objectives. We consider 2 models with multimodal input i.e. image+language pretraining and 3 models with predictive processing i.e. next-word prediction objective where only LLaVa-OneVison-7B combines both i.e. it is a multimodal model with predictive processing objective.

For the text-based models (Llama-3-8B-Instruct and CodeLlama-7B), we use the caption data provided in the BMD. From a set of five given captions for each stimulus, we conducted a review of captions and chose the longest caption to ensure maximal information about the stimuli is preserved in the caption. For the image-based models (ResNet-50, ViViT-B, BLIP-L), we first extract 32 frames uniformly distributed across the duration of each video and then average the model representations across frames. For the video-based models (ViViT-B and LLaVA-OV-7B), we also input 32 frames uniformly distributed across the duration of each video.

For each model, we extracted three sets of representations while processing each stimulus: an early layer, an intermediate layer and the last layer. For the early layers, we extracted features from layers corresponding to a quarter of the total number of hidden state outputs for each model. The intermediate layers were extracted from the midpoint based on the number of hidden layers for each model, while the last layers were the last of the hidden layers in each model. The exact numbers of extracted layers are indicated in Table 1. For ResNet-50, the layers used in our analysis can be extracted by accessing each Sequential (PyTorch) layer, which is the block part of the ResNet model, and for all the other models, the layers can be extracted by accessing hidden_states (huggingface).

## 2.4 Representational dissimilarity matrices (RDM)

After extracting neural and model representations for the 102 video stimuli, we create RDMs to quantify how the stimuli were represented in each brain area and model. The RDMs enable the comparison of representation across systems. The dissimilarity between two representations $X_i$ and $X_j$ can be expressed as:

$$D(X_i, X_j) = 1 - \frac{\text{cov}(X_i, X_j)}{\sigma(X_i)\sigma(X_j)} \tag{1}$$

where $cov(X_i, X_j)$ denotes the covariance between the two representations, with $\sigma(X_i)$ being the standard deviation of $X_i$. This formula denotes a $1 - Pearson$ distance between the representations of video stimuli $i$ and $j$.

The RDM is a symmetric $nn$ matrix $R$ where $R_i j$ reflects the dissimilarity between the representations of stimulus $i$ and stimulus $j$, resulting in a 102 by 102 matrix in the current study. Mathematically, the RDM is given by:

$$R_{ij} = D(X_i, X_j) \quad \forall i, j \in \{1, 2, \ldots, n\} \tag{2}$$

For the ROI-based analysis, we calculate RDMs for each subject and each ROI. For the whole brain searchlight analysis, RDMs are extracted within a sphere of radius 4 voxels centered at each voxel across the whole brain, measuring the local neural representation pattern.

## 2.5 Representational similarity analysis (RSA)

Representational Similarity Analysis (RSA) provides a common framework to quantitatively compare representational geometries across different modalities, such as computational models and neuroimaging data [42]. This approach has been particularly valuable in studying how both artificial neural network models and the human brain process complex, naturalistic stimuli like spoken or written language, images, and videos [55–57].

To assess the similarity of neural and model representations, we calculate the $(Spearman's\ \rho)$ between RDMs:

$$\rho = \frac{\text{cov}(\text{rank}(\text{vec}(R_A)), \text{rank}(\text{vec}(R_B)))}{\sigma(\text{rank}(\text{vec}(R_A)))\sigma(\text{rank}(\text{vec}(R_B)))} \tag{3}$$

$$\rho = \text{Spearman}(\text{vec}(R_A), \text{vec}(R_B)) \tag{4}$$

We correlate neural RDMs with the RDMs of the early, middle and late layers of models to measure how well model representations capture brain responses to stimuli.

We compute the upper noise ceiling as subject-to-group RDM correlation and the lower noise ceiling as leave-one-out RDM correlation per ROI or searchlight[58]. For multivariate reliability, the RSA values are normalized by the upper noise ceiling for both analyses for each subject, as described by:

$$\rho_i^{\text{norm}} = \frac{\rho_i}{\rho_i^{\text{upper}}} \tag{5}$$

Afterwards, we average the normalized correlations across the 10 subjects at each ROI or searchlight. The final corrected RSA value $\rho_i^{corrected}$ is given by:

$$\rho_i^{\text{corrected}} = \frac{\rho_i^{\text{norm}}}{N} \tag{6}$$

To statistically assess the searchlight RSA results, we follow Lahner et al. [41] and compute a one-sample two-sided t-test at each searchlight testing whether correlations differed from 0. The resulting p-values are FDR-corrected across searchlights (assuming positive correlation, $q = 0.05$). For the ROI-based RSA, a one-way ANOVA was performed at each ROI level using noise-normalized correlation for all 6 models (Bonferroni corrected with $n = 20$ ROIs, $p < 0.05$) and a Tukey's Honestly Significant Difference (HSD) test was performed as a post-hoc test for significant ROIs (FEWR $= 0.05$). A schematic diagram of the RSA procedure is shown in Figure 1B.

# 3 Results

## 3.1 ROI-based RSA

We compare the similarity of model representations extracted from different layers to fMRI activations for corresponding stimuli across different brain regions, as shown in Figure 2.

For the representations in the early layers, we see that:

- Most transformer architectures perform better than the baseline `ResNet-50`, with `ViViT-B` showing highest alignment for the early visual regions (V1v, V1d, V2v, V2d) while `LLaVA-OV-7B` was more aligned across other ROIs. This could indicate that for higher-level brain regions, multiple input modalities lead to better alignment.

- The `BLIP-L` model shows near-comparable performance to `LLaVA-OV-7B` in the early visual ROIs, with the performance gap increasing in other brain regions. This suggests that over

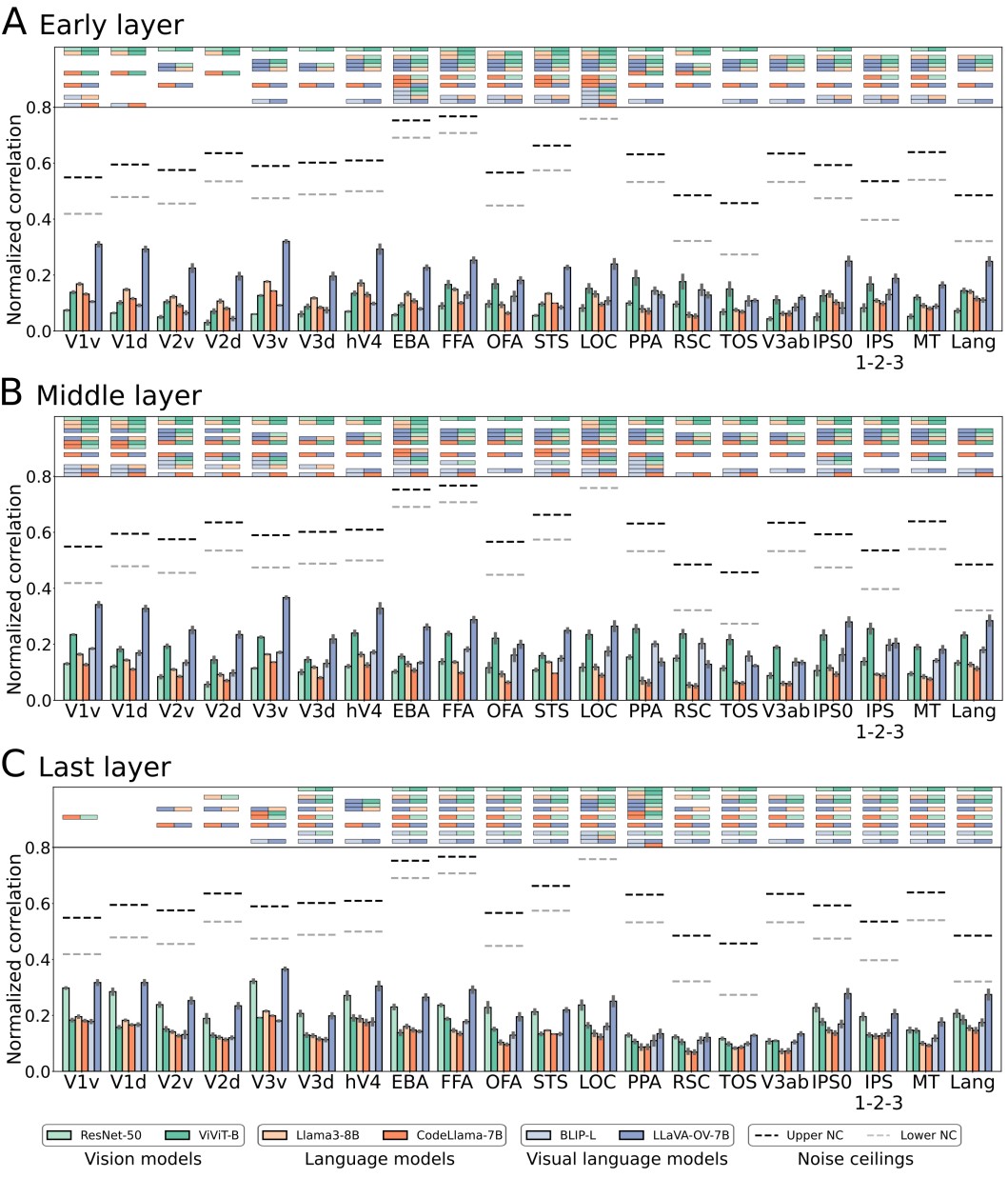

Figure 2: ROI-based RSA results. Noise-normalized Spearman's correlation coefficient values for (**A**) the early layer of each model, (**B**) the middle layer of each model, and (**C**) the last layer of each model. Error bars indicate the ± standard error. Noise normalization is done by using the upper bound of the noise ceiling for each ROI. Noise ceilings are shown for each ROI (lower bound: gray line; upper bound: black line). A one-way ANOVA test compared the noise-normalized correlation between all 6 models for each ROI (Bonferroni corrected, n = 20, $p < 0.05$). If significant, a Tukey's HSD test identified pairwise significance (FWER=0.05; significant pairs marked by dual paired color bars on top of each ROI plot).

and above input modality, the training objective also impacts performance, with improved alignment for predictive processing models.

- Text-based LMs (`Llama-3-8B-Instruct` and `CodeLlama-7B`) perform the worst in early and mid-level visual regions but show improvement in higher-level visual areas and dorsal, MT, Lang regions. This suggests that predictive processing leads to broader cognitive alignment but multiple input modalities tend to further improve performance (as seen by the better alignment of `LLaVA-OV-7B`).

- The `CodeLlama-7B` model mostly lags behind its natural language counterpart, `Llama-3-8B-Instruct`, suggesting that training with code, as opposed to only natural language, might lead to lower brain alignment even though their reasoning performance improves [24, 25]

For the mid-layer representations, we see that:

- The middle layer of the baseline `ResNet-50` achieves higher brain alignment compared to its early layer representations. The alignment of middle layer transformer representations is approximately similar to their early layers, with `ViViT-B` best modelling early visual regions (V1v, V1d, V2v, V2d, and V3d), and `LLaVA-OV-7B` showing high similarity to brain activations in other ROIs.

- For `BLIP-L`, the alignment trend is similar to that seen in its early layer, although the middle layer performs better in early visual ROIs. However, `LLaVA-OV-7B` still outperforms `BLIP-L`, supporting our conclusion that a predictive processing objective leads to better alignment.

- For the text-based LMs, the middle layer results are similar to the early layers, with `Llama-3-8B-Instruct` still outperforming `CodeLlama-7B`.

For the last layer representations of the model, we see some noteworthy patterns of alteration in alignment measures:

- The performance of `ResNet-50` is better compared to most other transformers in the early visual areas (which replicates some earlier findings [32]) but `LLaVA-OV-7B` still remains the best-performing model across all ROIs. Surprisingly, however, the alignment for `ViViT-B` and `BLIP-L` drops for early visual ROIs. This could indicate the importance of a predictive processing objective for retaining representational nuances across model layers.

- The performance of text-based LMs also remains similar or shows slight improvements across all ROIs, yet `LLaVA-OV-7B` still outperforms the text-based models. This highlights the superiority of multimodal models even within the predictive processing model class.

Moreover, we find that the representations from all our 6 models are similarly well-aligned not only to modality-specific regions (early visual areas or the language network), but also to regions involved in higher cognitive functions like IPSO or IPS1-2-3. This further strengthens the view that model representations also capture information in regions beyond low-level sensory processing areas.

## 3.2 Searchlight RSA

The whole-brain searchlight analysis is designed to provide a more detailed understanding of "where" in the human brain specific local response patterns exhibit similarity to the way the model encodes and represents the videos. By systematically evaluating neural activity across the entire brain, this method enables the identification of precise regions that align with the model's representational structure. The findings from the ROI-based (Region of Interest) analysis, which focused on predefined brain areas, were further validated and strengthened by the searchlight approach, offering a more comprehensive, data-driven confirmation of the model-brain alignment across broader cortical regions, as shown in Figure 3.

For the representations in the early layers, we see that:

- Image-based models `ViViT-B` and `RestNet-50` exhibit diverse alignment across different brain regions while `ViViT-B` shows better alignment for early to middle visual areas.

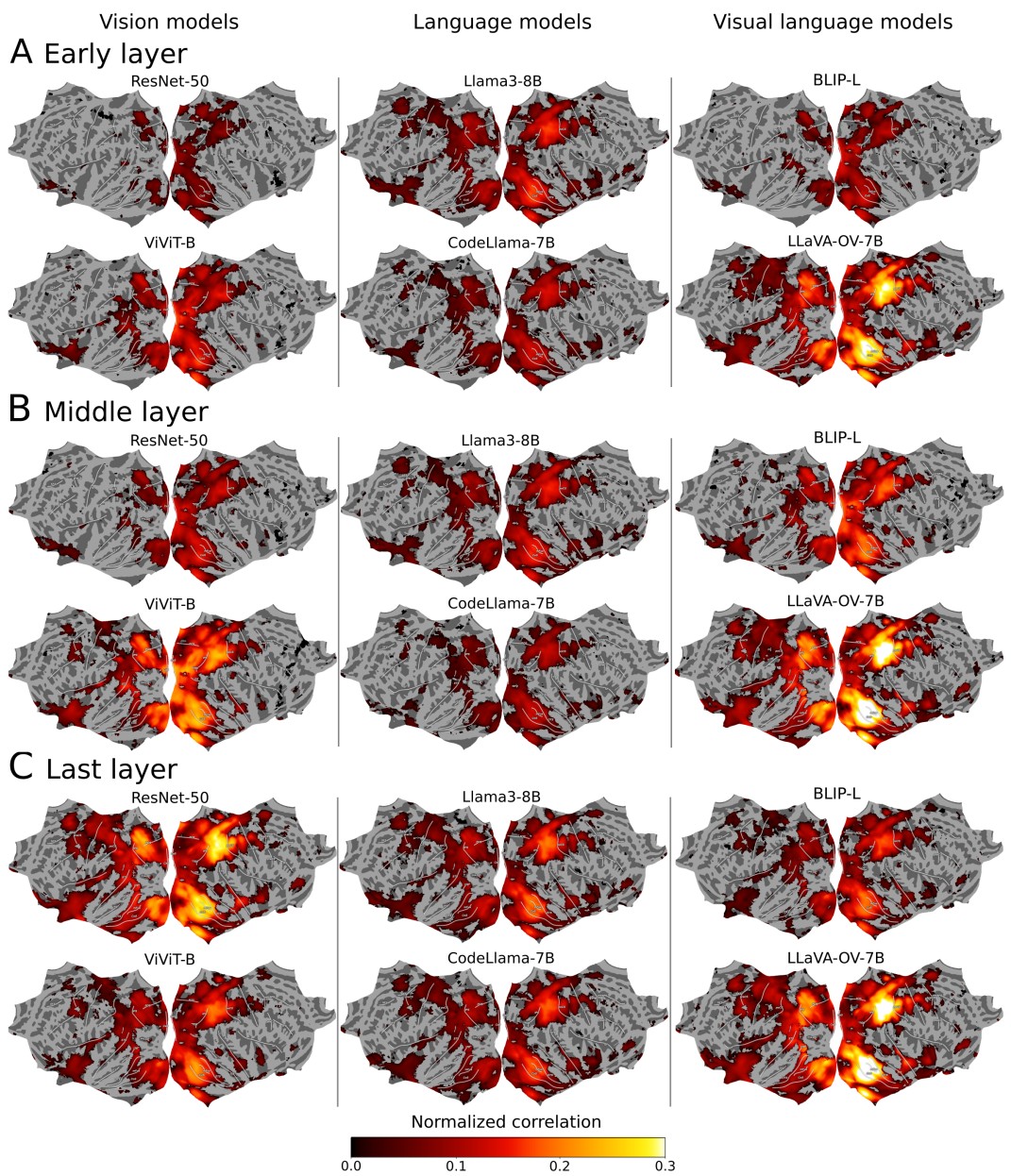

Figure 3: Searchlight RSA results. Noise-normalized Spearman's correlation coefficient values for (**A**) the early layer of each model, (**B**) the middle layer of each model, and (**C**) the last layer of each model. From left to right, the columns show non-language, language-only, and multimodal models. Noise normalization was done by using the upper bound of the noise ceiling at each searchlight. FDR correction ($q = 0.05$) was performed across searchlights.

- For video+text based model `LLaVA-OV-7B`, the alignment performance is the best among all six models. Though this model does not reflect the activation pattern of early visual areas, it aligns well with brain regions for middle to high-level visual processing and cognitive control functions.

This shows that while transformer architectures with predictive processing are more brain-aligned, enhancement with visual input modalities can further improve the alignment of representations.

In case of the middle layer representations, the trends seems to indicate that training with image data, image-based model `ViViT-B` and image+text-based model `BLIP-L` show stronger alignment compared to the early layer. Meanwhile, they exhibit strong alignment with superior parietal areas, which are related to higher cognitive functions such as attention modulation and multimodal sensory integration.

For the last layer representations of models, the trends are as follows:

- Similar to ROI-based analysis, the alignment for image-based model `ViViT-B` and image+text-based model `BLIP-L` drops for all the voxels that their the middle layer reflect, especially for early visual ROIs.
- Baseline image-based model `RestNet-50` gained stronger alignment overall compared to the middle layer, especially the early visual ROIs and abstract information selective ROIs.
- The video-text based model `LLaVA-OV-7B` still performed the best in brain alignment, remaining the strong mapping to the brain areas that reflects by middle layer.

Similar to the ROI-based analysis, the searchlight analysis reveals that the transformer architecture yields the highest alignments when combined with a next-word prediction objective and multimodal training data.

## 4 Discussion

The findings of this study offer valuable insights into the factors that shape the alignment of transformer-based models with neural activity in the human brain. Our results emphasize the significance of architectural design, training strategies, and task modalities in influencing the representational overlap between artificial models and biological systems.

**Impact of Training Data and Multimodal Learning** While transformers with predictive processing objectives (`Llama-3-8B-Instruct`, `CodeLlama-7B` and `LLaVA-OV-7B`) demonstrate strong representational alignment with human brain regions, our findings suggest that the training data also plays a critical role in improving this alignment. The `CodeLlama-7B` model mostly lags behind its natural language counterpart, `Llama-3-8B-Instruct`, suggesting that training with code, as opposed to only natural language, might narrow model's ability to capture natural language patterns and features as effectively. Additionally, models trained on multiple input modalities, such as both vision and language, tend to exhibit stronger and more generalized alignment with brain activity (`LLaVA-OV-7B`). This multimodal training likely enables models to capture richer, more integrated representations that better reflect the multifaceted nature of human cognition, where various sensory and cognitive inputs are processed simultaneously.

The human brain continuously integrates information from different sources—visual, auditory, linguistic, and more—during perception and decision-making. By training transformers on both vision and language, for example, the models are able to capture complex interactions between these modalities, resulting in a more holistic and cognitively aligned representation that resonate more with how the brain processes diverse inputs [59]. This is particularly evident in non-linguistic brain regions where models trained solely on language show weaker alignment, whereas multimodal-trained models achieve a more robust alignment across diverse cognitive networks. This suggests that incorporating multimodal data into training protocols enhances representational alignment capabilities.

**Impact of Training Objective** The training objective seems to play a crucial role in determining brain alignment. Transformer-based models like `ViViT-B` and `BLIP-L` show worsening alignment with early visual areas as we pass through model layers, which can be attributed to their alternate training objectives such as classification or image captioning. On the other hand, the models with next-word prediction objectives such as `Llama-3-8B-Instruct`, `CodeLlama-7B` and `LLaVA-OV-7B` all improve and retain alignment even at the last layer representations. This suggests that predictive processing objectives might better reflect cognitive mechanisms in the brain [60, 61], also evidenced by studies of the human visual system's higher-level areas providing predictive signals into lower regions to "explain away" things [62–65]. These results could also count as supporting evidence of the human visual system as a predictive machine. Moreover, the increasing alignment observed in higher layers of the models may indicate that more abstract, high-level representations in the brain and models converge when predictive objectives are used.

These findings highlight the potential of predictive processing frameworks not only to improve the performance of artificial models but also to enhance their ability to simulate human cognitive processes. The success of these models in aligning with human brain activity may encourage further research into training paradigms that prioritize prediction as a core objective.

**Broad-scope Cognitive Alignment** Our findings demonstrate that the representations from all six models not only maintain alignment trends with modality-specific regions (such as early visual areas and the language network) but also extend to regions associated with higher-order cognitive functions, including IPSO, IPS1-2-3, and superior parietal areas, which aligns with previous research [66]. This supports the notion that model representations are capable of aligning with brain regions beyond low-level sensory processing, offering insights into more complex neural dynamics.

Moreover, this broad-scope alignment suggests that the models are not confined to superficial sensory features but are capable of capturing abstract cognitive processes, suggesting that they might share an abstract representation space beyond respective modalities.

**Conclusion** Our study raises critical questions about the nature of representational capabilities in transformer LMs- particularly as a function of their training data and objectives. The results of this study have broader implications for both neuroscience and artificial intelligence research. In the field of AI, the findings provide valuable insights into how architectural and training choices impact a model's ability to mimic human cognitive processes. Our results suggest that the combination of multimodal training and predictive processing objectives may be particularly effective in developing models that align with human neural patterns, opening new possibilities for creating more cognitively aligned artificial systems. From a neuroscientific perspective, transformer architectures may offer new opportunities to explore how the brain processes information across various domains. Their ability to align with low to high-level cognitive brain regions suggests that these models can be useful tools for studying the neural basis of cognition more broadly.

# 5   Limitations

Despite the promising findings, our study has a few limitations that must be considered when interpreting the results. First, the analysis primarily focuses on visual processing and language comprehension tasks, limiting the generalisability of our results to other cognitive domains. Cognitive processes such as memory, attention, and abstract reasoning are not directly explored, leaving open the question of how well transformer-based models align with brain regions involved in these tasks. Future studies should incorporate a broader range of cognitive tasks to determine whether the representational alignment observed here extends to other domains of cognition.

Secondly, our study only tests a selection of transformer models. To validate our findings and ensure the robustness of representational alignment, future research should include a wider range of models to assess how generalizable our results are and whether certain model architectures or training paradigms consistently produce stronger alignment with neural activity across multiple cognitive domains. This approach would provide a deeper understanding of which specific model features contribute most to effective alignment with the human brain.

Finally, while we demonstrate an alignment of models with brain regions, it is important to recognize that alignment does not necessarily imply equivalence in cognitive mechanisms. The models we analyzed are optimized for prediction and generation in artificial tasks, and their internal representations may not map directly onto biological processes in a straightforward way. Thus, further work is needed to establish a deeper understanding of how these models function as proxies for brain activity.

## Acknowledgments and Disclosure of Funding

The authors would like to express their sincere gratitude to Neuromatch Academy for providing an outstanding platform for learning, collaboration, and research. The online course and project facilitated by Neuromatch Academy not only brought the authors together but also provided valuable opportunities for presenting earlier versions of this work and receiving critical feedback from the broader community. We would also like to thank our mentors, colleagues, and peers for their insightful comments and support throughout the research process.

The work was supported by a Vector Institute Research Grant and an Natural Sciences and Engineering Research Council of Canada Discovery Grant to Y.M.

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
