# OpenReview forum: "Investigating the role of modality and training objective on representational alignment between transformers and the brain"
_NeurIPS.cc/2024/Workshop/UniReps — UniReps_

### Official Review · Reviewer_pjHo · 2024-10-06
**Investigating the role of modality and training objective on representational alignment between transformers and the brain**

**Rating:** 5
**Confidence:** 5

**Review:**

The paper presents interesting work  executed with sound methodology.
However, the discussion moves too swiftly to conclusions without providing sufficient depth or critical analysis and I feel it's a bit immature.

Regarding model selection, some choices are puzzling. For example, while the inclusion of ResNets, vision transformers, language models, and multimodal models is well-justified, the use of CodeLlama seems out of place. The reasoning behind selecting CodeLlama is unclear, especially given that key models widely referenced in the literature, such as CLIP (and its video counterpart, XCLIP), are missing. A significant portion of the vision-decoding literature—by authors like Takagi, Ferrante, Scotti, Ozcelik, VanRullen, Bencherith, Chen, Liu, and many others—relies on CLIP or related models, making its absence here noteworthy.

Additionally, the paper overlooks crucial literature on brain-language alignment. Seminal works from Gallant’s group on semantic representations, visual-language mapping studies from Huth’s lab, and Jean-Rémi King’s work on model-brain alignment are conspicuously absent. This omission limits the context in which the results are interpreted.

The topic of representational alignment between multimodal models and brain activity has been explored previously, often with findings that multimodal models better predict neural activity. While this study uses a unique dataset (videos), it’s crucial to contextualize these results within the existing body of research to avoid simply reiterating established findings. Works such as Choksi et al. (2022), Conwell et al. (2021), The Algonauts Project (2023), and Antonello et al. (2024) provide examples of studies that would be relevant to this discussion.

The selection of models also raises questions about the choice to focus primarily on visual representations. Given the multimodal nature of the stimuli (videos), why not also consider audio models? Addressing such omissions could enhance the depth of the analysis.

The main conclusion—that multimodal models generally outperform single-modality models in explaining brain activity—is in line with previous findings. However, the study could benefit from exploring deeper questions: Does model task performance correlate with representational alignment to the brain? Do larger models align better with neural activity, or is it more about the quality or size of their embeddings?

Integrating these broader considerations and situating the findings within the existing literature would enhance the overall contribution of the paper, so I would suggest the authors to improve their work.

However I'm perfectly aware that this is a workshop and a work in progress could stimulate discussion and be of interesting for the community, so I don't have anything against seeing this work at the workshop since the contribution is well done and solid.

Minimal references that should be discussed and included:

Multimodal neural networks better explain multivoxel patterns in the hippocampus, Choksi et al 2022.

What can 1.8 billion regressions tell us about the pressures shaping
high-level visual representation in brains and machines?  Conwell et al 2021,

What can 5.17 billion regression fits tell us about
artificial models of the human visual system?

The Algonauts Project 2023 Challenge: How the Human Brain Makes Sense of Natural Scenes A. T. Gifford et al 2023

How Many Bytes Can You Take Out Of Brain-To-Text Decoding? Antonello et al 2024

---

> ### Author Response · Authors · 2024-10-28
>
> We thank reviewer pjHo for their comments. We address the comments raised by the reviewer below:
> - Thank you for your insightful feedback. We are excited to expand this work further and will incorporate CLIP in the future extension of our study. The questions raised by the reviewer are valuable, and we look forward to addressing them in our upcoming studies. We see this as an opportunity to strengthen our work, and we are grateful for your constructive input.
> - The reason why we used CodeLlama in this work is that code-trained language models showed different behaviours in some reasoning tasks (Dhar and Søgaard, 2024). We have clarified this in the camera-ready version (3rd paragraph on Introduction).
> - We have excluded models that have audio as their modality because the dataset we used did not include audio in their stimuli. We have included this information in the camera-ready version (1st paragraph on Stimuli).
> - Thank you for pointing out the missing references. We acknowledge the oversight and have revised the manuscript accordingly. We have now included the relevant papers suggested by the reviewer in the Introduction and Discussion sections of the camera-ready version. After careful consideration, we determined that the study by Antonello et al. (2024) is not directly relevant to our work, as it focuses on brain-to-text decoding, which differs from the scope of our research. Therefore, we have chosen not to include it in our references.

---

### Official Review · Reviewer_o558 · 2024-10-06
**This work studies a variety of transformer models to determine what effects different training objective may have on brain-model similarity, with thorough experiments and interesting results.**

**Rating:** 8
**Confidence:** 3

**Review:**

Synopsis:

The authors investigate the brain-model alignment of multiple LLMs / VLMs. They find different degrees of alignment between the models they studied, which were of a variety of different types w.r.t. input modality and training objective. They also find that different models tend to do better with different regions of interest in the brain, including some higher level, more abstract ROIs that are not directly related to specific input modalities (ex: vision models intuitively do better on V1, but perhaps not on theory of mind ROIs). The differences between these models' alignment points towards the degree to which different input modalities and training objectives may increase brain-model alignment.

Pros:

- This paper does well to select a variety of models, especially some that are around the same size. Sometimes it can be difficult to tell if brain-model alignment is just better because a model is larger and more expressive, but this way the impact of the input modality / training objective is more apparent. Particularly the Llama / LlaVA models all seem to be around 7-8 billion. However I'm not familiar with ViViT-B or BLIP-L. I think that having one extra column in table 1 with the number of trainable parameters would have been a nice touch.

- To the best of my knowledge, I have not seen work that has studied brain-model alignment using video input to calculate similarity. Much prior work has studied text and image modalities alone. While this work only studies six different models, this is still a nice step towards that direction and provides interesting results.

- Somewhat similarly, I also have not seen much use of searchlight RSA techniques. This was a good idea and it was nice to see corroboration of your results through a different technique as well.


Cons:

- There are some things which could be spelled out quite a bit more explicitly so that the paper is more reproducible by others. A more minor note is which layers you chose. I understand that your early and middle choices correspond to the quarter and halfway layers of the models, but what if there is an odd number of layers and these don't line up exactly? Since there are only six models, this could just be mentioned explicitly. More importantly, which part of the layer do you take? Transformer blocks have the attention mechanism and then MLPs as well. Most of what I've seen uses the MLP layers to do this type of analysis. If you had come up with a nice way to derive similarity from the attention mechanism as well that would have been good, but even if just using the MLP portion, saying that explicitly is best.


Questions:

- Specifically which layers do you take out of each model, and which parts of the layers?


Overall:

Overall I like this work. I haven't seen an analysis of different LLMs / VLMs before, especially w.r.t. their input modalities or training objectives. The results show interesting trends that can help lead future work in the area, and seems to be especially relevant given the recent growth of multimodal models in general. Helping to show the use of these models as encoding models in neuroscience is good work - I recommend acceptance.

---

> ### Author Response · Authors · 2024-10-28
>
> We thank reviewer o558 for their feedback and insightful comments.
> - We have included the number of parameters of each model in Table 1 of the camera-ready version.
> - We have made an edit to clarify what exact layers were used for each model in the camera-ready version (Table 1 and 3rd paragraph on Model representations).

---

### Official Review · Reviewer_8YS9 · 2024-10-07
**deconstructs representational alignment across transformer architectures and specific brain regions**

**Rating:** 8
**Confidence:** 3

**Review:**

This paper takes a deeper look at representational alignment across transformer-based language models and the human brain. A matrix of alignments is examined between language models trained on various modalities, training data, and objectives against specific brain regions involved in various cognitive functions. It presents a thorough examination of not only the trade off on various types of multimodal models for alignment to brain activity, but cross references those against 20 specific brain regions (ROIs) to differentiate alignment between brain locations responsible for vision or language. I am most familiar with alignment in the context of neural networks, and here they follow sound methodology such as including Resnet-50 as an interesting baseline and separating out the alignment effects of models trained for predictive processing. Overall I felt this paper was very relevant to the themes of the workshop and offers a more through examination of human/machine alignment than I had previously found in the literature.

---

> ### Author Response · Authors · 2024-10-28
>
> We thank reviewer 8YS9 for their detailed review and valuable comments.

---

### Official Review · Reviewer_ZSfQ · 2024-10-07
**Ambitious scope, but more controlled comparison of inductive biases is required to derive meaningful conclusions**

**Rating:** 4
**Confidence:** 4

**Review:**

The authors attempt a fairly ambitious sweep of model-brain comparisons using large-scale fMRI data to study the impact of modality and training objective on representational alignment. I applaud the scope of the project, and the particular question of how different modalities interact in these contexts is extremely relevant right now. I also appreciate that the authors attempted to discuss limitations of their work in the Discussion.

However, the paper suffers from a common but serious pitfall of comparing models that differ in many respects, e.g., simultaneously varying combinations of training task, dataset, and architecture. This makes it impossible to truly understand which inductive bias underlies better brain alignment when differences show up in the analyses.

For these sorts of comparisons, to make any claim about the impact of one inductive bias vs. another, a more systematic approach to model comparison is warranted. In particular, it is critical to hold all inductive biases constant and only vary one at a time. This paper provides a roadmap for such comparisons:
https://www.biorxiv.org/content/10.1101/2022.03.28.485868v2
and similar issues are discussed here:
https://www.biorxiv.org/content/10.1101/2022.09.27.508760v2

In short, controlled comparisons of different inductive biases are necessary for studies of this nature to provide meaningful theoretical advances. For this reason, unfortunately, I feel that further revision is required to warrant acceptance. Figure 2 is hard to parse—key results should be summarized and presented in a more concise way. Similarly, Figure 3 is too coarse to derive meaningful conclusions about anatomical distinctions in brain predictivity.

Two other methodological suggestions: There was unclear justification for capping the number of voxels per ROI at 1000—a reliability-based thresholding would be substantially more appropriate. Finally, the method of choosing layers to analyze was arbitrary, and a cross-validation procedure to identify the most predictive layer of each model per brain region would be more appropriate.

---

> ### Author Response · Authors · 2024-10-28
>
> We thank reviewer ZSfQ for their comments. We address the comments raised by the reviewer below:
> - Thank you for your valuable feedback. We will use a better systematic approach for the model comparison in our future work, referring to the suggested references.
> - Capping the number of voxels per ROI to 1000 was done by the authors of the BMD paper (Lahner et al., 2024), which is the dataset we've used in this work. The authors of BMD did the reliability-based thresholding and chose the top 1000 voxels based on the t-contrast probability map. We have incorporated this information in the camera-ready version (2nd paragraph on fMRI representations)
> - Thank you for your insightful suggestion. We will use a cross-validation approach in our future work. We have made an edit to clarify which layers were used for each model in the camera-ready version (Table 1 and 3rd paragraph on Model representations).

---

### Decision · Program_Chairs · 2024-10-10

**Decision:**

Accept

**Comment:**

In light of the positive reviewers' feedback and relevancy of the submission, we are pleased to accept this paper for presentation at UniReps 2024. We kindly ask the authors to incorporate the reviewers' suggestions and feedback in the final camera-ready version of the manuscript, especially in the way results are grouped by variation factors.